# Untangling the Context-Specificity of Essential Genes by Means of Machine Learning: A Constructive Experience

**DOI:** 10.3390/biom14010018

**Published:** 2023-12-22

**Authors:** Maurizio Giordano, Emanuele Falbo, Lucia Maddalena, Marina Piccirillo, Ilaria Granata

**Affiliations:** Institute for High-Performance Computing and Networking (ICAR), National Research Council (CNR), V. Pietro Castellino 111, 80131 Naples, Italy; emanuele.falbo@icar.cnr.it (E.F.); lucia.maddalena@cnr.it (L.M.); marina.piccirillo@icar.cnr.it (M.P.); ilaria.granata@cnr.it (I.G.)

**Keywords:** essential genes, machine learning, graph neural network

## Abstract

Gene essentiality is a genetic concept crucial for a comprehensive understanding of life and evolution. In the last decade, many essential genes (EGs) have been determined using different experimental and computational approaches, and this information has been used to reduce the genomes of model organisms. A growing amount of evidence highlights that essentiality is a property that depends on the context. Because of their importance in vital biological processes, recognising context-specific EGs (csEGs) could help for identifying new potential pharmacological targets and to improve precision therapeutics. Since most of the computational procedures proposed to identify and predict EGs neglect their context-specificity, we focused on this aspect, providing a theoretical and experimental overview of the literature, data and computational methods dedicated to recognising csEGs. To this end, we adapted existing computational methods to exploit a specific context (the kidney tissue) and experimented with four different prediction methods using the labels provided by four different identification approaches. The considerations derived from the analysis of the obtained results, confirmed and validated also by further experiments for a different tissue context, provide the reader with guidance on exploiting existing tools for achieving csEGs identification and prediction.

## 1. Introduction

The concept of gene essentiality was first introduced in the context of synthetic biology and microbial engineering as a key topic related to the definition of the minimum set of genes essential for allowing life, known as the minimal genome [1]. In particular, essential genes (EGs) were considered those that cannot be removed or silenced from a genome without provoking a deleterious phenotype, reducing the organism’s viability or fitness [2]. Thanks to technological progress, EG investigation moved from microorganisms to more complex organisms, such as mice and worms, up to humans. Then, gene essentiality acquired a wider role, becoming a key concept of genetics, with implications ranging from basic research to evolutionary, systems biology, and precision medicine [3]. The experimental strategies to define how necessary a gene is for the organism’s survival or fitness consist of knocking out the gene and examining the consequent phenotype. Gene knockout (KO) can be achieved through different methodologies, such as transposon mutagenesis, targeted RNA interference, and CRISPR-Cas9, with the latter considered the state-of-the-art method for its higher efficiency and precision [4]. In the case of humans, the gene-editing techniques are clearly performed on cell lines in vitro, and EGs are considered those genes necessary for maintaining cell growth, proliferation, and survival [5]. Most cellular EGs encode conservative functional elements, which contribute mainly to DNA replication, gene translation, gene transcription, and intracellular and extracellular transport [6]. An attempt to identify the human EGs in vivo is represented by the estimation of gene mutation scores from population-level data of sequencing experiments, according to the assumption that EGs rarely carry disrupting mutations [7]. Considering the involvement of EGs in fundamental biological processes, it is quite clear how their study is crucial in many fields, such as genetics, industrial biotechnology, medicine, and evolutionary biology, both from a basic and an applied research point of view.

The development of gene-editing approaches coupled with next-generation sequencing for genome-wide loss-of-function screening allowed the identification of EGs in thousands of cell lines. Big international projects, such as Project Achilles (Broad Institute, [8]), Project DRIVE (Novartis [9]) and Project Score (Sanger Institute [10]), contributed to furnishing data on a large panel of human cell lines from multiple cancer lineages to create a catalogue of EGs [2]. The massive amount of data generated unravelled the conditional nature of essentiality, according to which a gene can change its status depending on the genetic and environmental context [11]. Generally, we can distinguish common EGs, also defined as core fitness genes (CFGs), which are indispensable in all the contexts considered and for which there are no compensatory mechanisms to allow survival following gene inactivation, and context-specific EGs (csEGs), whose essentiality is manifested only in certain conditions. Because the experiments are mainly performed on tumour cell lines, a large part of the EG research is devoted to the cancer context. The EGs provide a clear definition of the requirements for sustaining the basic cell activities of individual human tumour cell types and can represent key targets for cancer therapies, especially if they show conditional essentiality. Given the richness of data produced and shared with the scientific community by the abovementioned big collaborative projects, computational approaches to support the experimental procedures and to allow easier scalability are strongly needed. The results of gene deletion experiments, together with gene attributes characterising the essentiality, permit the development of computational methods to identify and predict CFGs and csEGs. Underestimating the context-specific essentiality can lead to a strong misinterpretation of EGs, as well as a lack of important opportunities for precision medicine. Although this concept is widely accepted by the genetic and biological community, the computational procedures proposed to identify EGs seem to ignore this aspect, focusing only on organism-wide EG prediction approaches. In recent years, different databases have been developed to facilitate the exploration of human EGs by providing data and gene annotation. To name a few: the Online Gene Essentiality (OGEE) database [3], which provides datasets of gene deletion experiments, lists of organism- and tissue-specific EGs, as well as tools to explore the data from large- and small-scale studies; the Database of Essential Genes (DEG15) [12], which contains information on all EGs that are currently available and built-in analysis modules by which users can perform various analyses, sub-cellular localisation distribution, gene ontology, and KEGG pathway enrichment analysis, and generation of Venn diagrams to compare and contrast gene sets between experiments; the Dependency Map (DepMap) [13], which is not strictly an EGs database, but rather a portal collecting cancer dependency experiments and data from several projects [8,9,10], also providing tools for visualising data and pan-cancer (common) EGs lists.

To emphasise the importance of considering the context-specificity when approaching the computational identification or prediction of EGs, here, we review the properties common to all EGs (Section 2) and those related to context-specific essentiality (Section 3), as well as the computational approaches for identifying and predicting EGs (Section 4). In an extended experimental section (Section 5), recently published machine learning (ML) and deep learning (DL) approaches have been adapted for predicting csEGs in the context of human tissues. To this extent, four different sets of labels, derived from three EG identification methods and OGEE, have been exploited to consider as many sources of annotation as possible. Our results allowed the discussion about the efficiency and the flexibility of the computational methods, the labelling approaches, and the choice of gene attributes for achieving a good prediction of csEGs.

## 2. Common Properties of Essential Genes

It has been estimated that EGs represent around 10% of the genome in humans, with most genes being not essential, likely due to compensation effects induced by redundant or alternative pathways and functions. In general, it is possible to affirm that larger genomes seem to require fewer vital genes to support the survival of species, according to a sort of economic strategy [14]. Identifying or predicting EGs requires the recognition of characteristics that allow distinguishing them from non-essential genes (nEGs). Depending on the need to discriminate CFGs or csEGs, these attributes can be generic or related to the condition of interest. Chen et al. [15] conducted a comprehensive study of human EGs, to investigate and collect their genomic, epigenetic, proteomic, evolutionary and embryonic characteristics. They also provided a web platform, the Human Essential Genes Interactive Analysis Platform (http://sysomics.com/HEGIAP, accessed on 23 May 2023), to explore the data and analyse a user-defined list of genes. Some of the most studied EGs properties are reported in Table 1, and, for ease of discussion, we subdivided them into eight main groups concerning different contexts.

EGs show common properties that contribute to their structural stability. Measures of DNA and protein stability are the gene/protein length, the guanosine-cytosine (GC) content, the count of transcripts generated by alternative splicing, and sequence-derived features that allow converting the sequence properties into numerical attributes, such as codon frequency, maximum relative synonymous codon usage (RSCUmax), codon adaptation index (CAI) and amino acid frequency. EGs are unexpectedly short and contain a variety of transcripts larger than nEGs [15]. These two characteristics of essential genes may be important for the stability of cell functions. EGs tend to be located in the most active chromatin regions and consequently are highly expressed and hubs of epigenetic modifications. The gene or protein expression can be considered the main context-specific property as it can refer to the condition of interest, such as a tissue, a cell type, a disease, or a certain environment. Function and localisation information are crucial. As may be expected, EGs show a biological centrality being involved in many crucial functions, processes, and pathways, such as those related to central metabolism regulation, DNA replication, protein translation, and cellular structure stabilisation. Furthermore, many studies have highlighted the important contribution of sub-cellular localisation in the prediction of EGs [16,17]. Given their crucial role, EGs are particularly conserved between species and over time. Furthermore, they show recognisable patterns in differential expression analysis of human diseases, particularly cancer. Disease-associated genes are likely intermediates between highly essential and non-essential genes [18]. In the context of cancer, EGs strongly overlap with cancer-driver genes, in which genetic mutations that cause abnormal and uncontrolled cellular growth occur with the highest frequency. EG transcription is well regulated in embryonic development and lineage segregation, especially for the development of the fetal origin part of the placenta [19]. Along with the biological and genetic features that characterise the EGs, there is a set of information that can be obtained from the interactions and behaviour of biological factors described through network structures. In the context of EG features, the protein–protein interaction (PPI) network, describing the physical connections among proteins, is the most exploited one. According to what is called the centrality–lethality rule, the greater the centrality of a gene, or its product, in a PPI network, the greater its essentiality [20].

## 3. Context-Specific Essentiality

The separation of genes into two sharply distinct categories, essential and non-essential, appears to be somewhat too simplistic. The scientific community has now widely accepted that many EGs are substantially context-dependent. Several recent papers addressing the properties of EGs [15] and the concept of essentiality [14] highlight that the latter is more a dynamic rather than a fixed intrinsic property and that it strictly depends on the genetic and environmental context. The attribute “essential” can further be extended to functions giving more strength to this concept, as functions are dependent on the cell type, tissue and/or response to the environment [21]. Essential cellular functions are highly conserved among species, and the components of a signalling cascade, pathway, or protein complex can be replaced by functional equivalents with an independent evolutionary origin, according to what is called non-orthologous gene displacement [14]. Moreover, as the characteristics defining essentiality, and also the results obtained from the gene deletion experiments, are treated as quantitative traits, it is clear that what makes a gene essential or not in a specific context is a threshold that can shift changing the context and the data considered. The essentiality score is established by assigning to each gene targeted by the knockout strategy, such as CRISPR-Cas9, a numerical value representing the fitness of that gene for cell viability [22]. This score might be highly dependent on the genotype, transcriptome, lineage of the cell, and environmental factors. In this scenario, genes can be essential in all, almost all, few or only one condition under investigation. This leads to the distinction between the CFGs, which are indispensable for cell survival regardless of context, and the csEGs. Hart et al. showed that CFGs are mainly involved in fundamental cellular mechanisms such as RNA splicing, DNA replication, and translation [23], and are particularly conserved between species [24]. The identification of csEGs is particularly appealing in the context of cancer research since they represent ideal candidate drug targets. In this perspective, in [25] the authors developed a strategy based on analysis and integration of genomic alterations data, signalling pathways, protein–protein interactome network, protein expression, dependency maps in cell lines and patient-derived xenografts, to reveal breast cancer EGs. In [26], the authors defined the genome-scale context-dependent essential genes in human pluripotent stem cells and showed that there is a growth substrate-dependent gene essentiality in cancer-derived cells. Analyses of context-specific essentiality through pan-cancer loss of function genomic cell line screens have shown a significant enrichment of synthetic lethality interactions, defined as lethality caused by simultaneous alteration of gene pairs that are otherwise individually viable [27]. If two EGs share the same profile, they are likely to be involved in similar functions [28]. The environmental dependence of essentiality is highlighted by mechanisms such as auxotrophy and antagonistic pleiotropy. The first refers to metabolic genes that are essential only if they encode elements absent from the natural environment and necessary for cell/organism survival. Similarly, antagonistic pleiotropy is a phenomenon for which gene deletion can be beneficial in some environments but deleterious in others. Byars et al. derived non-experimental evidence for antagonistic pleiotropy in many diseases, including cancer, neurodegenerative diseases, cardiovascular disease, and diabetes [29]. The intra- and inter-variability of diseases is the underlying main issue of precision medicine, which requires new strategies to predict and determine therapeutic responses in subgroups of patients stratified on the basis of specific genetic profiles [30,31]. This approach improves the effectiveness and safety of new treatments by limiting side effects and sparing patients from exposure to drugs that are unlikely to be successful or that are even dangerous. Considering that EGs are necessary for an organism or cell to survive in a specific context, it is not surprising that they can be potential targets for the development of new drug therapies. Furthermore, the evolvability of gene essentiality can help in understanding and predicting resistance to drugs [14]. As the increasing number of cell line assays has expanded the understanding of EGs in many new contexts, there has also been a concomitant effort to rationalise and integrate the work conducted by different groups. In this direction, two major contributions come from the DepMap and Project Score that offers a wide range of data and tools to explore both CFGs and csEGs [13,32].

## 4. Computational Approaches to Define Gene Essentiality

As previously mentioned, nowadays, the gene-editing technique mostly used for generating knock-out cells and identifying EGs with high selectivity is CRISPR-Cas9 (hereinafter referred to as CRISPR), although other methodologies can be applied, such as RNA-interfering, homologous recombination, and transcription activator-like effector nuclease. From knockout experiments, scores can be derived that reflect the cell response to deletion of the specific target gene. Consequently, the labelling of a gene as essential strictly depends on the setting of a threshold for the experimental scores.

Computational approaches are then strongly needed to support the experimental process and objectively define the threshold that divides essential from non-essential genes, as well as to allow increasing the scale of the investigation to genome-wide and to multiple conditions (e.g., cell lines), since the experimental procedures are cost- and time-laborious. With respect to their goals, we can divide the computational methods adopted into two main categories, which we call (i) identification and (ii) prediction methods. The identification methods refer to the assignment of the labels “E” (“Essential”) or “NE” (“Not Essential”) to genes by thresholding the scores derived from gene-editing experiments (bottom left of Figure 1), and can be achieved through knowledge- or data-driven approaches. The prediction methods, instead, refer to machine learning approaches that, exploiting gene attributes related to the essentiality property, allow to compute for each gene the probability of being essential or not (bottom right of Figure 1). The attributes mainly come from omics science and network biology. Most of these methods are supervised and exploit the labels “E”/“NE” assigned to the genes through the identification methods, and in some cases collected in dedicated databases.

### 4.1. Identification Methods

The CRISPR knockout system has been derived from a mechanism of bacterial immunity. Briefly, it is based on targeting the gene sequence of interest through a single-guide RNA (sgRNA) that leads the Cas9 nuclease to generate a specific DNA double-strand break. The cell mechanism of nonhomologous end-joining pathways is then activated to repair the DNA damage generating insertions or deletions which ultimately lead to gene knockout. Finally, next-generation sequencing is performed to analyse the sgRNA abundance in cells. Through computational methods, the resulting reads are mapped to the original sgRNA library and counted. By exploiting various approaches, the raw data can be processed and hit genes identified. Exhaustive recent reviews about the computational methods to analyse CRISPR screen data are provided by [33,34]. Although many methods have been developed, the estimation of gene fitness from CRISPR screens remains a not trivial task. Some pitfalls need to be taken into consideration, such as (i) the lack of replicates can lead to a large variance of the model; (ii) the read count distributions might be different depending on the screen libraries, and therefore the normalisation of the data is crucial; (iii) different thresholds can be set to distinguish between EGs and nEGs from the final scores, and thus different sets of EGs can be observed even when starting from the same initial data set. The methods can be divided into those that accept as input the raw data from the sequencing task and furnish a complete workflow till the identification of EGs, and those that, instead, have been developed to analyse the pre-processed fold-changes (fc) data, as provided, for example, into the DepMap portal. Among those belonging to the first category, we mention MAGeCK, one of the first and most used tools to address the processing of CRISPR sequencing raw data [35]. After a median normalisation of sgRNA read counts, MAGeCK computes a binomial distribution to test the difference significance between two conditions (e.g., the cells undergoing the KO experiment and the control group). The p-values of the distribution are used to rank sgRNAs, and the robust ranking aggregation (RRA) algorithm [36] identifies positive and negative enriched genes, and thus their fitness. Another widely employed method is BAGEL [37], which uses a gold standard reference set of EGs and nEGs for the training phase. The model firstly estimates the fc of sgRNAs from the training sets; then, the likelihood of the sampled fc of sgRNAs of the targeted gene is computed from the essential or non-essential training set. The final outcome is a logarithm Bayes factor (BF), which reflects the gene fitness; the more negative the BF, the more essential the gene.

Regarding the methods accepting pre-processed data in the shape of fc matrices, Hart et al. developed the Daisy model (DM) to identify CFGs from genetic screens of multiple cancer cell lines [23,38]. The pipeline assumes that input data are log2(fc) observations from biological replicate experiments of gene-editing screens. The fitness of each gene for all screened cell lines is conceptually represented by a petal of a daisy, and its overlap with all other petals constitutes a common set of genes (the core of the daisy), identified as the CFG set. Following this idea, De Lucia et al. proposed a generalisation of DM, the adaptive Daisy model (ADaM) [39], to overcome the limitation of an a priori arbitrarily defined minimal number of cell lines in which a gene has to be essential to be defined as CFG. The ADaM method includes a semi-supervised algorithm that adaptively computes the minimal number of cell lines in which the KO of a gene should cause a reduction in viability for that gene to be considered an essential gene. By considering genes as elements of non-fuzzy sets, i.e., the cell lines, the ADaM method calculates the fuzzy intersection of these sets by adaptively determining the number of non-fuzzy sets the gene must belong to be considered an element of the fuzzy intersection as well (the membership threshold). This threshold maximises the difference between the expected and actual cardinality of the resulting fuzzy intersection (the final set of EGs), as well as the coverage of predefined elements.

The CoRe R package v 1.0.0 (https://github.com/DepMap-Analytics/CoRe, accessed on 31 May 2023) implementing ADaM also provides an additional unsupervised approach, named fitness percentile (FiPer) [39]. The authors propose four variants of this model; a final output can be obtained by intersecting the results of these four, i.e., the FiPer consensus model. In contrast to ADaM, FiPer starts from the scores matrix and sorts all cell lines in decreasing order based on their dependency on the gene under investigation; then, the rank position of the gene is extracted by sorting all the genes by their fitness effect on tested cell lines. Afterwards, the density of these scores is estimated using a Gaussian kernel function, and the genes whose scores fall below the minimum of the density plot are defined as essential. This approach is entirely independent of a reference list of EGs. For both ADaM and FiPer approaches, depending on the set of cell lines taken into account, we can identify CFGs or csEGs. As observed by the authors, ADaM is more stringent than FiPer in terms of the cardinality of the E class, and this stringency influences their use for identifying CFGs or csEGS.

Another unsupervised approach is given by CEN-tools, which is an integrated database and set of computational tools to explore context-specific gene essentiality from pooled CRISPR data sets [40]. Briefly, CEN-tools starts from the corrected log(fc) matrix computed using the MAGeCK pipeline after CRISPRcleanR correction from the Project Score analysis pipeline [35] and builds a logistic regression model that is trained to separate essential and non-essential genes. Then, the probabilities of each EG and nEG across all cell lines are sorted in increasing order and converted into a discrete range frequency of probabilities, which generates a matrix of probabilities. Finally, a clustering procedure applied to such a matrix of probabilities identifies the set of most essential genes, separating CFGs from context-specific essentials [40].

The methods described so far can be considered as data-driven approaches applying statistical distribution on gene-editing screen data. Another way of deriving thresholds to divide EGs from nEGs is by using a knowledge-driven approach. Motivated by the results in [15], we employed this approach in [41,42] to provide EG labels subsequently used for predictive experiments. In particular, the CRISPR scores (CS) were divided into eleven groups from CS0 to CS10, and the labels vector was obtained by assigning the label of the gene to the most frequent score group among the cell lines. In particular, the genes having the most negative values of gene effect scores were grouped in the CS0 class, while the remaining genes from the union of groups CS*x*–CS9, with *x* varying from 1 to 7, were considered non-essential. For each gene, CS10 was not frequent enough across cell lines to become the ultimate representative label. In order to include all the genes and provide a comparison with the other identification methods, here, we divided the CS0 group from the union of all remaining groups CS1–CS9 and refer to the method hereinafter as CS0–CSX.

### 4.2. Predictive Models

Recently, several reviews have been devoted to computational methods for the prediction of EGs [42,43,44,45,46]. A concise summary highlighting the taxonomies proposed and the period covered by the reviewed literature is given in Table 2.

Most computational methods exploit network topology-based features, generally extracted from PPI networks. As previously mentioned, node centrality measures are correlated with the essentiality property. Various types of topology measures, based on neighbourhood, path, or eigenvector information, or their combination, are taken into account [42,47]. Also, network embedding [41,42,48,49,50,51,52] is frequently adopted to exploit the underlying network information, encoded into a low-dimensional space.

To improve the accuracy of prediction, PPI network-based information is frequently integrated with information coming from other types of networks, such as co-expression [47] or metabolic [41] networks, and/or from different biological sources, including genomic, transcriptomic, epigenetic, functional, and evolutionary characteristics [41,47,49,50,51,52,53].

The methods we adopted for the experiments, chosen based on the availability of the related software, are all ML-based methods (with three of them specifically based on DL models), exploiting not only the topology of the PPI networks but also additional biological information.

The prediction method we proposed in [42], in the following named GrEGs, is based on the light gradient boosting machine (LGBM) [54], where the ensemble of decision trees is trained in a supervised mode using as targets the gene labels obtained by the CS0–CSX identification method (see Section 5.1). The input to the model ensemble is a set of gene attributes, ranging from a large set of genetic and functional characteristics, to PPI networks topological features learned and represented as vectors in a multidimensional real space by means of the node2vec [55] graph embedding technique.

Recently, Zhang et al. [50] proposed DeepHE, a method based on DL models, to predict human essential genes by integrating characteristics derived from both sequence data and PPI network. The experimental results obtained showed that DeepHE can accurately predict human gene essentiality with an area below the ROC curve (AUC) above 94%.

Schapke et al. [53] presented EPGAT, an approach for predicting essentiality using Graph Attention Networks (GATs). EPGAT utilises graph neural networks (GNNs) to capture node relationships in the PPI network automatically. The authors also incorporated, one at a time, three biological features (gene expression profiles, orthology information, subcellular localisation) during the model learning process. The performance of EPGAT has been evaluated on *S. cerevisiae*, *E. coli*, *D. melanogaster*, and *H. sapiens*, comparing its ROC AUC values with other commonly used network-based and ML techniques. EPGAT demonstrated performance similar to node2vec embedding but with a shorter training time.

In their study, Kuang et al. [51] developed XGEP (expression-based gene essentiality prediction), an ML approach aimed at predicting the essentiality of both protein-coding genes and long non-coding RNAs in cancer cells. They employed a collaborative embedding method applied to transcriptomic profiles from the TCGA project [56]. The relevant biological information for the learning task was extracted through three different methods: collaborative embedding [57], Gene2Vec [58], and autoencoder [59]. Subsequently, they used the feature vectors obtained from the collaborative embedding to construct models based on gradient-boosted trees, support vector machines, and deep neural networks (DNN) to predict EGs.

## 5. Experimental Study on Context-Specific EGs Identification and Prediction

In this section, we report our experience in applying some EG identification and prediction methods in the context of kidney tissue. As a prerequisite for our investigation, we only explore identification and predictive methods with freely available implementation software, adapting, where possible, the input data to the context under investigation. For the label identification methods, we also considered the public repository OGEE from which we could gather the kidney-specific EG labels.

We first must point out some important issues regarding our experimental use of both EG identification and prediction methods: (1) the extraction and pre-processing of input data (gene attributes and CRISPR score data) related to the kidney-specific domain was carried out by following the procedures described in the reference works and no other data processes were taken into account; (2) all methods were applied following the parameter configurations indicated as optimal in the reference works, and no parameter optimisation was performed in the tissue-specific domain considered for the current experimental study; (3) although from the experiments related to the investigation on prediction models for EGs we collected performance metrics, it is important to highlight that the third-party implementations of the methods used for the investigation adopt different validation techniques.

Taking into account the above considerations, the scope of this experimental study was providing a guide on how to exploit the tools in the literature for achieving csEGs identification and prediction rather than presenting a performance comparison of the different methods. However, as a natural consequence of this work, we could derive and discuss some considerations about the approaches used and the results obtained. We exploited these considerations to select some of the methods and confirm and validate the results obtained on a second tissue context.

### 5.1. Identification of Human-Kidney-Specific CFGs

As mentioned in Section 4.1, the identification of EGs and nEGs from knock-out screens are often dependent on the setting of a threshold, and, depending on this and on the method applied, different sets of EGs could be obtained. For this reason, we explored several methods and analysed their overlapping. In particular, three of the approaches presented in Section 4.1 has been exploited to identify the EGs from CRISPR-Cas9 screening across kidney-specific human cell lines: CS0–CSX, which can be defined parametric, as it requires the setting of knowledge-driven thresholds; AdAM, which can be considered semi-parametric, as it exploits a set of gold-standard EGs to discriminate the genes; and finally the FiPer consensus model (hereafter referred to simply as FiPer), that is totally independent of thresholds but for this reason less stringent. The three identification methods have been applied to the latest CRISPR knockout gene effect score data (DepMap v. 23Q2; https://depmap.org/portal, accessed on 31 May 2023), downloaded from the DepMap portal, accounting for 17,453 genes and 32 kidney cell lines. Negative scores imply cell growth inhibition and/or death following the gene knockout; thus, the more negative the score, the more essential its corresponding gene is.

Furthermore, a pre-compiled list of kidney-specific EGs has been downloaded from the OGEE database. In its current version (OGEE v3) [3], it contains experimentally tested essential and non-essential genes for 91 non-human species, of which 16 eukaryotes and 75 prokaryotes. CRISPR and RNAi experimental data of 931 cell lines from 27 human tissues are collected (as reported on the web page), with lists of EGs divided into pan-cancer and tissue-specific EGs.

By comparing the labels obtained using the three identification methods and the pre-compiled list of EGs (Figure 2), it can be observed that, as expected, the FiPer method returned the highest number of EGs (E = 1409), followed by ADaM (E = 1303), CS0–CSX (E = 889), and OGEE (E = 885) (Figure 2). The authors of the CoRe methods suggested a different application for them according to the different stringency given by the number of EGs. In this case, it is worth noting a reduction in the difference between the two, with 1409 and 1303 EGs identified by FiPer and ADaM, respectively, likely due to the narrow context under study and the smaller dataset of experimental scores used. The lower number of EGs obtained from CS0–CSX and OGEE makes them more stringent with respect to the other approaches.

Figure 2 shows the overlapping of kidney-specific EGs as identified by the four approaches considered. In total, 682 genes were shared among all four sets of EGs, suggesting more reliability in considering this core set of genes as essential for the context under investigation A significant overlap can be observed between FiPer and ADaM, with the first covering almost all the genes identified by ADaM (57 genes are not present in FiPer). In total, 155 genes were identified solely by FiPer and 20 by ADaM. OGEE and CS0–CSX gave a comparable number of EGs identified (885 and 889, respectively), very close to the number of genes shared among all the sets (682), but there are 710 genes overlapping between them, highlighting some differences. The OGEE list includes 36 genes not shared with the others, while only 5 are unique for CS0–CSX. The stringent nature of the CS0–CSX approach can encourage higher confidence in the identification of csEGs. However, it must be pointed out that the CS0–CSX approach is data-driven and its generalisation to different data needs to be tested.

### 5.2. Prediction of Human-Kidney-Specific CFGs from Multiomics Data

In this section, we report an experimental study aimed at predicting csEGs, by using a selected set of methods described in Section 4.2. For this analysis, we adopted as target labels of kidney-specific EGs those obtained through FiPer, ADaM and CS0–CSX approaches, as well as those provided by the OGEE database, as described in Section 4.1.

As for input data, several types of gene attributes were exploited, following the approach proposed by the authors of each method:Features extracted from the DNA sequences of the genes under investigation (named “seq” in Table A1, Table A4, Table A5, Table A6, Table A7, Table A8, Table A9, Table A10 and Table A11). The sequence features were obtained through the R packages seqinr v. 4.2-16 [60] and protr vs. 1.6-3 [61];Features extracted from the PPI network by means of node2vec (named “embn2v<size>” in Table A4, Table A5, Table A6, Table A7, Table A8, Table A9, Table A10 and Table A11). The kidney-specific PPI was downloaded from the Integrated Interaction Database, which provides networks with comprehensive tissue, disease, cellular localisation and druggability annotations [62]. The tissue-specificity was obtained by filtering the edges by their tissue annotation;Features extracted from the correlation of TCGA transcriptomic data by means of collaborative embedding (named “embcf<size>” in Table A4, Table A5, Table A6, Table A7, Table A8, Table A9, Table A10 and Table A11). Expression data of the three subtypes of renal cancer (KIRP, KIRC and KICH) were downloaded from the GDC portal (https://portal.gdc.cancer.gov, accessed on 30 May 2023). Data were processed as described in [51] before being submitted to the collaborative embedding procedure;Gene expression values of kidney healthy samples (named “exp” in Table A1, Table A4, Table A5, Table A6, Table A7, Table A8, Table A9, Table A10 and Table A11) as gathered and used in Section 2 of [42]. Data were downloaded from the GTEx portal (https://gtexportal.org, accessed on 22 May 2023);Orthology and sublocation features of the genes (named “orth” and “subloc” in Table A1, Table A4, Table A5, Table A6, Table A7, Table A8, Table A9, Table A10 and Table A11) as described and provided by [53];Biological attributes including genetic and functional characteristics collected, as in Section 2 of [42], using generic and tissue-specific biological information (named “bio” in Table A1, Table A4, Table A5, Table A6, Table A7, Table A8, Table A9, Table A10 and Table A11).

In detail, we have selected the four EGs prediction methods for which the software was freely available and described in Section 4.2: EPGAT [53], XGEP [51], DeepHE [50] and GrEGs [42]. In the experiments, we applied these methods under the parameter settings indicated as optimal in the related works, summarised in Table A3.

DeepHE uses the features listed in Table A1 as “seq”, i.e., the information extracted from the DNA sequences of the genes being tested. This type of attribute clearly is not context-specific. In addition, the “embed” features were produced by applying the node2vec embedding scheme on the kidney PPI.

Since XGEP uses differential expression features extracted from the TCGA transcriptome data through a collaborative embedding technique [57], the data from three subtypes of renal cancer were considered (KIRP, KIRC, and KICH), separately and together. Once the gene embeddings have been computed, they were used as input to the DNN models to predict gene essentiality in cancer cells. The kind of input used makes it particularly focused on the prediction of cancer-specific EGs. Clearly, the input data, consisting of gene co-expression data, can be adapted to each context for which expression data are available, but in the view of not modifying the scope of the tool as thought by the authors and also to give an example of disease-specific EGs prediction we used the TCGA data of renal cancer subtypes.

EPGAT exploits the information automatically extracted (learned) from the PPI through the GCN: following the authors’ guidelines, we concatenated it with orthology, subcellular localisation and gene expression data (“orth”, “subloc” and “exp” in Table A1), taken alone or all together, to represent the nodes in the input layer of the GCN. During the training, the GCN updates the representation of each node on the next layer by incorporating information from a broader neighbourhood of nodes.

For GrEGs, the attributes are genetic characteristics (“bio”+“exp” in Table A1) and PPI embedding features (“embn2v64” in Table A1), taken alone or combined. In the experiments, we also considered a validation procedure similar to the one adopted in DeepHE, i.e., applied to a sub-sampling of the original dataset such that the ratio E:NE is 1:4, so as to test DeepHE and GrEGs on the same sets of features (Table A8, Table A9, Table A10 and Table A11).

### 5.3. Analysis of Prediction Methods for Human-Kidney-Specific CFGs

Figure 3 reports the performance measures (mean and standard deviation) obtained by the validation procedure of each model built by training it (in a supervised mode) on each of the four labelling strategies. Here, we report only the measures corresponding to the cases in which all gene feature sets were used as input. Table A4, Table A5, Table A6, Table A7, Table A8, Table A9, Table A10 and Table A11 report all experimental results, including the performance obtained through the different models exploiting the feature sets alone or in combinations.

As previously mentioned, the scope of this experimental work was not comparing the performance of the different methods; nonetheless, from the obtained results, we can derive some important considerations.

Regarding the metrics, the ROC-AUC slightly varies throughout the experiments and methods, demonstrating that it is not suitable to appreciate the differences in performance in the case of binary classification of strongly unbalanced classes. As is known, accuracy is not a reliable metric to represent performance when dealing with unbalanced problems either, as it does not provide indications about the prediction accuracy of each individual class.

Regarding the methods, the poor performance obtained with XGEP makes us hypothesise a miscorrelation between the labels, obtained from in vitro experiments on kidney cancer cell lines, and the gene expression data from tissue samples of renal cancer patients. The best performance indeed was obtained considering all the cancer subtypes data (“embcf150_PAN_RenalCancer”) and with the less stringent labelling strategies (ADaM and FiPer), namely when considering the largest panel of input data and "E" labels. The authors already declared that the performance of XGEP decreases in the case of cancer-type-specific expression data compared to the whole pan-cancer dataset. They suggest the need for more expression profiles, but in our opinion, an improvement in the performance could probably be achieved by deriving labels from mutation scores of the corresponding cancer data.

In the case of EPGAT, the best results were shared by the “PPI+ortho” and “PPI+subloc” combinations, depending on the metric considered, partially confirming what the authors found when using human data, which is the best ROC-AUC in the case of “PPI+subloc” data. We also exploited all the annotation sets jointly (“PPI+exp+ortho+subloc”), but no improvement was achieved.

Looking at the results obtained through DeepHE and GrEGs, important considerations can be derived about gene attributes and their contribution to the prediction performance. Regarding these methods, which use the node2vec embedding on the PPI, it seems that the embedding data give the greatest contribution to the discrimination capability of the predictors (see the BA and MCC metrics of Table A4, Table A5, Table A6, Table A7, Table A8, Table A9, Table A10 and Table A11). This is proved by the fact that using the “seq” and “bio+exp” attributes alone determines a poor performance compared to the cases in which the embedding attributes are used alone (“embn2v64”) and the cases where they are used in conjunction (“seq+embn2v64” and “bio+exp+embn2v64”). Furthermore, all but the model adopted for GrEGs are based on DL: EPGAT adopts a GNN to process gene input features in a learning process following the PPI topology, while the others apply a DNN as an ML classifier on input tabular gene data. Only in the case of DeepHE, part of the data table is represented by vector-like embeddings produced by the linearisation of the PPI network. This is likely the reason why DeepHE, by using node2vec embedding, seems to be the best performing among DL methods. When the GrEGs model, which is not based on DL, is cross-validated on an undersampled input dataset, in the manner implemented by DeepHE, it showed similar (even better) results (Table A8, Table A9, Table A10 and Table A11). This evidence seems to suggest that the method is not decisive for performance. No matter if a traditional ML method or a DL model is used, the discriminative power is likely due to the embedding extracted from the PPI.

Having validated the GrEGs model on an undersampled dataset in the same way as the authors of DeepHE, we were able to compare the two methods by interchanging the input data to derive further considerations on the attributes (see Table A8, Table A9, Table A10 and Table A11). The “bio+exp” features of GrEGs, given as input to DeepHE, determined a notable performance improvement compared to the “seq” attributes, likely due to the fact that the latter are not context-specific at all. The opposite, i.e., using the “seq” attributes instead of the “bio+exp” ones with GrEGs, confirmed this consideration since they lead to a performance decrease. In combination with PPI embedding features, it seems that, especially for GrEGs, adding “bio+exp” to “embn2v64” features (“bio+exp+embn2v64”) determined a greater improvement than adding the “seq” ones (“seq+embn2v64”). Joining together all the features in the combination “seq+bio+exp+embn2v64” determined a slight decrease for DeepHE, while better performance for GrEGs. The more features are used, the better performance is achieved in the case of GrEGs, except in the case of OGEE labelling.

### 5.4. Testing a Different Context: Identification and Prediction of Human Lung-Specific CFGs

Taking advantage of the results and considerations made on the kidney tissue so far, we applied some of the methods to a second tissue case study: the lung. In detail, we labelled the genes through ADaM and FiPer identification methods, as they can be considered the most generalisable methods among those used in this work. The identification has been applied to the latest CRISPR knockout gene effect score data (DepMap v. 23Q2; https://depmap.org/portal, accessed on 20 November 2023), downloaded from the DepMap portal, accounting for 17,453 genes and 113 Lung cell lines. Differently from the case of the kidney tissue, the two considered identification methods showed similar stringency. Indeed, the number of E genes identified by the two methods was comparable, 1650 for ADaM and 1647 for FiPer, with 1501 E genes in common (Figure 4). Regarding the prediction methods, we selected the ones that exploit the PPI embedding, as it gave the strongest contribution to the classification task. The classification performance results obtained with DeepHE and GrEGs on ADaM and FiPer labels are shown in Figure 5 and Table A12 and Table A13. The results confirmed what we observed previously: the best performance was achieved by the GrEGs method when it was cross-validated on an undersampled input dataset, as implemented by DeepHE, and when all the features were combined (“bio+exp+embn2v64”).

### 5.5. Performance Evaluation on csEGs and CFGs

Identifying the csEGs automatically implies that the set of genes marked as E includes both those common to other contexts and those that are specific to the context under study. In order to evaluate the performance on the csEGs, we subtracted the CFGs from the predicted EGs. The set of CFGs was downloaded from OGEE, where they are named core essential genes (CEGs) and defined as genes essential to 80% or more tested cell lines, including 606 genes. For both kidney and lung datasets, csEGs were considered those labelled as E and not present in the CFGs set. The true positive rate (TPR) and false negative rate (FPR) were calculated for both csEGs and CFGs as the percentage of elements correctly predicted and wrongly assigned to the negative class, respectively. Figure 6, showing the four calculated rates, highlights a very good performance on CFGs with TPR values ≥80% and a good but improvable performance on csEGs. The best results in terms of csEGs prediction were achieved by using ADaM labelling on lung data (TPR = 71.33%).

## 6. Concluding Remarks

The use of computational models aimed at identifying EGs has received an increasing boost because of the technical advances in experimental procedures that allow the generation of massive amounts of data. Thus, the development of dedicated computational approaches is urgently needed to deal with scalability, time and economic issues.

Considering the essentiality as a dynamic, evolvable and condition-specific property is of fundamental importance in the view of the precision medicine goals. Indeed, the discrimination of csEGs and CFGs allows for identifying candidate biomarkers and elucidating biological processes. Although this crucial issue is widely accepted, the computational methods proposed by the recent literature are mostly focused on organism-wide EGs. Depending on the input data and the goal to pursue, we considered the computational approaches as belonging to two main categories that we called (i) identification and (ii) prediction methods. With the aim of providing a small guide for those who want to approach the identification and/or prediction of csEG, we presented the current work in which we: first introduced the topic by going through the recent literature concerning the main characteristics of EGs, their context dependency and the computational approaches developed; then, we illustrated an experimental study of the applicability of these latter to the context-specificity domain, reporting some considerations on the outcome of this experience. In particular, extensive experimental work has been applied to the kidney-specific context and, taking advantage of the results obtained, a selected set of methods has been applied to a second tissue context. In the light of reproducibility and open science, we also made all the data used for the experiments available (https://www.kaggle.com/datasets/cdsgroupicarcnr/cseg-biomoleculesmdpi-dataset, accessed on 20 November 2023).

Four identification approaches have been used to produce lists of genes labelled “E” or “NE”: ADaM, FiPer, CS0–CSX and OGEE. The latter refers to labels retrieved from the relative database, whereas the other three embed proper analytical and computational schemes for EG identification from KO experimental scores. Intersecting the EGs obtained through the four approaches, we found a great overlap among all (682 genes). The lowest amounts of EGs were those obtained from OGEE and CS0–CSX, suggesting a higher stringency and probably higher confidence in identifying csEGs. However, between the two, CS0–CSX was the one with the highest amount of EGs shared with all the others. It is worth noting that OGEE and CS0–CSX, deriving from a database and from a knowledge-driven approach, respectively, are hard to adapt to different user-defined contexts.

As an efficient predictive model requires both consistent labels and data, and the annotations produced by the identification methods can also serve as gene labels to train ML- or DL-supervised models for EGs prediction. Four methods, i.e., XGEP, EPGAT, DeepHE and GrEGs, have been tested to classify EGs complying with the parameters and gene attributes used by the authors. The most intriguing consideration that we derived from the experimental results was that the embedding techniques are able to automatically learn topological features from the biological networks that give the strongest contribution to the discriminative power, regardless of the classification method applied downstream. The crucial role of the PPI embedding features has been discussed in [41,42]. The importance of the correspondence between the context-specificity of attributes and labels was inferable by the results obtained with XGEP. We think indeed that the poor performance achieved was due to the fact that, while the labels come from in vitro experiments on tumoural cell lines, the input data are from renal cancer patients. As a confirmation of our hypothesis, a poor overlap between EGs identified using in vitro and in vivo (i.e., mutation metrics calculated on exome sequencing data) has already been demonstrated [7]. In this regard, it would be interesting to explore the usage of embedding features from single-cell transcriptomics data [63].

As we did not perform any parameter optimisation, a direct comparison of the methods cannot be discussed. Still, we demonstrated that by applying the cross-validation to GrEGs as in the manner proposed by DeepHE, namely undersampling the dataset, the performance improved, suggesting to put attention to the validation approach and to the unbalancing issue. Interchanging the attributes between these two methods allowed us to state that adding biological and genetic annotations to the PPI embedding features determined an improvement of the performance, especially in the case of using context-specific attributes, such as the “bio” ones, while the sequence derived attributes (i.e., “seq”) did not seem to contribute likely because not context-specific at all. In order to confirm and validate these considerations, we took advantage of the results obtained on the kidney tissue and applied some selected methods on a second tissue context. In particular, we labelled the genes through ADaM and FiPer, as they can be considered the most generalisable methods, and predicted those labels by using DeepHE and GrEGs, as we noticed that the features obtained from the PPI embedding gave a crucial contribution to the discrimination of genes. The results obtained on the lung tissue context confirmed what we observed previously on the kidney: making the two methods comparable by undersampling the input dataset in both cases, GrEGs with its full features combination, including general and tissue-specific attributes, gave the best performance. Furthermore, as we are interested in individuating csEGs, we calculated the prediction performance on CFGs and csEGs separately. The estimation of TPRs and FNRs suggested that the most appropriate identification method for csEGs was ADaM, as also stated by the authors, and that, as expected, the more data are available, as for the lungs, the better are the results. The classification performance was particularly good in the case of CFGs and good, but still improvable, for csEGs. Although we cannot suggest a method over another for the reasons explained above, as far as we can tell, this work represents the first attempt to furnish an overview, theoretical and experimental, of knowledge, data and methods devoted to the investigation of csEGs.

## Figures and Tables

**Figure 1 biomolecules-14-00018-f001:**
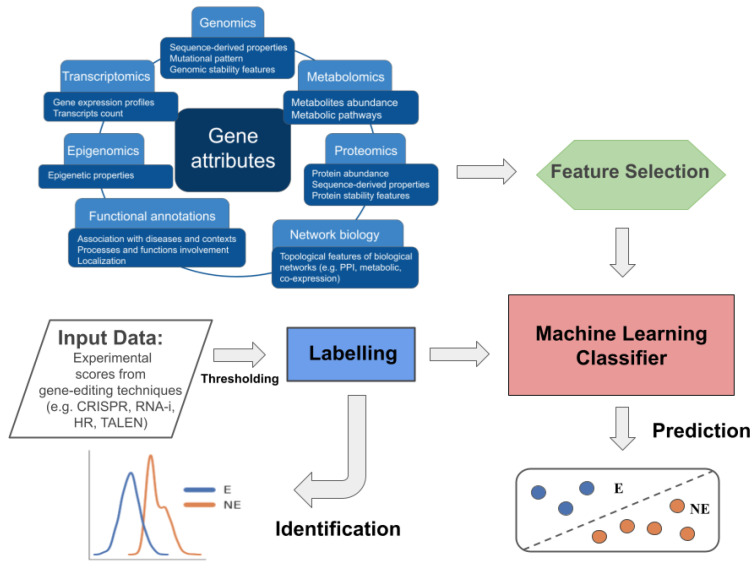
Schematic representation of the workflows characterising the identification and prediction of EGs. The identification process is conceptually shown in the bottom left part of the figure and involves computational methods to identify a threshold on experimental gene-editing screen data used to label genes as E or NE. The prediction, instead, requires two main inputs: (1) gene attributes characterising the essentiality, mostly regarding gene annotation and data from omics science and topological features from biological networks; (2) E/NE labels from dedicated databases or identification methods. Attributes are eventually subjected to a feature selection step, and labels, are then used in the supervised training of ML models for EG prediction.

**Figure 2 biomolecules-14-00018-f002:**
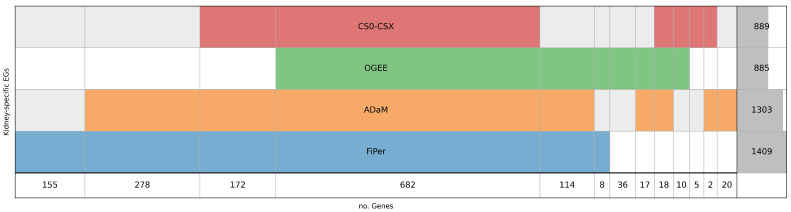
Diagram representing kidney-specific EGs intersection among CS0–CSX, OGEE, ADaM and FiPer. Each row represents a set of E genes. The last row reports the number resulting from the intersections or the number of genes not shared with the others for each set. The last column on the right indicates the total number of E genes included in each set, with the dark grey shadow representing the corresponding histogram.

**Figure 3 biomolecules-14-00018-f003:**
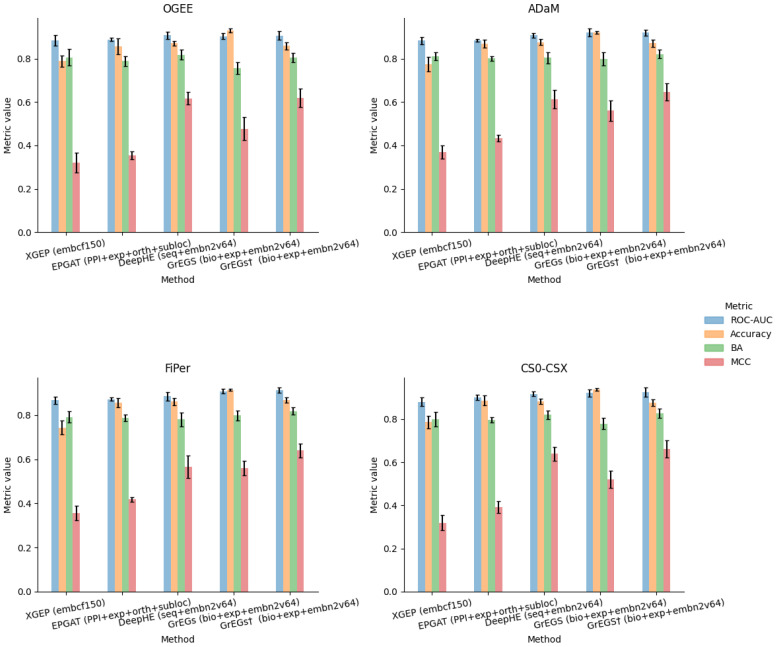
Plots of performance results for XGEP, EPGAT, DeepHE and GrEGs prediction models for kidney-specific EGs when applied to each labelling method and with full input data set.

**Figure 4 biomolecules-14-00018-f004:**
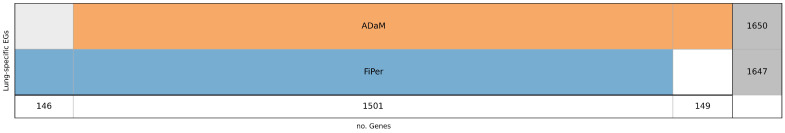
Diagram representing lung-specific EGs intersection among ADaM and FiPer. Each row represents a set of E genes. The last row reports the number resulting from the intersections or the number of genes not shared with the other for each set. The last column on the right indicates the total number of E genes included in each set, with the dark grey shadow representing the corresponding histogram.

**Figure 5 biomolecules-14-00018-f005:**
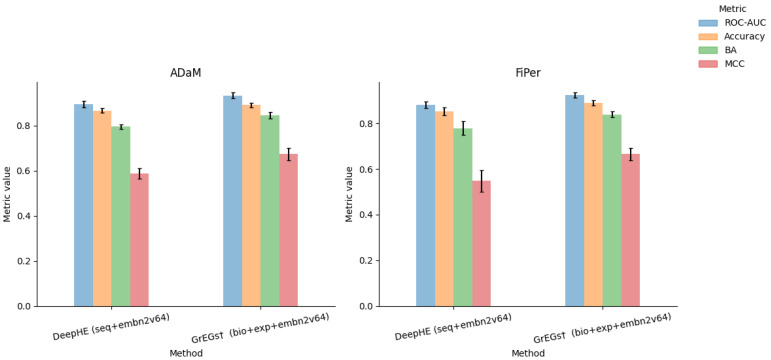
Plots of performance results for DeepHE and GrEGs prediction models for lung-specific EGs when applied to AdAM and FiPer labelling methods and with full input data set.

**Figure 6 biomolecules-14-00018-f006:**
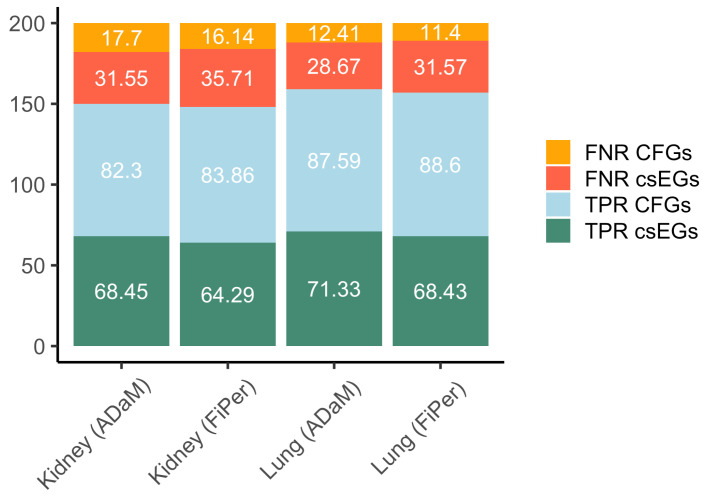
Stacked barplots showing the percentage of csEGs and CFGs (alias CEGs, as furnished by OGEE) correctly predicted (TPR) and not (FNR) considering as total the number of E labelled genes served as input for prediction methods for the kidney and lung cases.

**Table 1 biomolecules-14-00018-t001:** Genomic, transcriptomic, epigenetic, functional and evolutionary features of EGs.

Feature Context	Attributes
Structural stability	Gene/protein length
	GC content
	Transcripts count
	Gene/protein sequence-derived properties
Gene/protein expression	Transcripts abundance
	Protein abundance
Function and localisation	Functional annotation
	Pathway involvement
	Subcellular localisation
Epigenetics	Transcription factor binding
	Chromatin accessibility
	DNA methylation
	Histone modification
Conservation/evolution	Orthologs count
	Protein stability
	Evolutionary age
Association with disease	Gene–disease association
	Cancer driver mutation
	Differential expression
Embryonic development	Gene expression pattern
Network biology	Topological attributes

**Table 2 biomolecules-14-00018-t002:** Summary of recent reviews on computational methods for the prediction of EGs.

First Author	Years	Taxonomy
& Refs.	Covered	
Rasti [43]	2001–2017	(1) Network topology-based
		(2) Integrating PPINs and biological information - Sequence-based (sub-cellular localisation, evolutionary conservation, and gene expression) - Machine learning-based
Li [44]	1987–2018	(1) Network topology-based (exploiting neighbourhood, path, eigenvector information, or their combination)
		(2) Integrating PPINs and biological information
		(3) Dynamic network-based
		(4) Machine learning-based
Dong [45]	1996–2018	Modeling methods implementing/combining five types of features:
		(1) Evolutionary conservation
		(2) Domain information
		(3) Network topology
		(4) Sequence component
		(5) Expression level
Aromolaran [46]	2004–2021	(1) Homology mapping
		(2) Constraint-based
		(3) Machine learning-based (intrinsic/extrinsic features)
Granata [42]	2019–2021	(1) Network topology-based
		(2) Classical machine learning-based
		(3) Deep learning-based

## Data Availability

The data used for the experiments are available at https://www.kaggle.com/datasets/cdsgroupicarcnr/cseg-biomoleculesmdpi-dataset (accessed on 20 November 2023). For EG identification, the ADaM and FiPer methods are implemented in the CoRe package available at https://github.com/DepMap-Analytics/CoRe (accessed on 20 November 2023), while the approach underlying the CS0–CSX method is described in https://github.com/giordamaug/EG-identification---Data-Science-in-App-Springer (accessed on 20 November 2023). The implementations of EG prediction methods adopted in this work are available at: DeepHE—https://github.com/xzhang2016/DeepHE (accessed on 20 November 2023) (as GitHub archive); XGEP—https://github.com/BioDataLearning/XGEP (accessed on 15 October 2023) (as GitHub archive); EPGAT—https://github.com/JSchapke/essential-gene-detection (accessed on 15 October 2023) (as GitHub archive); GrEGs—https://www.kaggle.com/cdsgroupicarcnr/cseg-biomoleculesmdpi-gregs (accessed on 20 November 2023) (as runnable Python notebook).

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
