# Peer review of "Untangling the Context-Specificity of Essential Genes by Means of Machine Learning: A Constructive Experience"

_biomolecules, 2023, doi:10.3390/biom14010018_

Round 1
Reviewer 1 Report
Comments and Suggestions for Authors
The manuscript by Giordano et al delves into the concept of gene essentiality and highlights the importance of considering context-specific essential genes (csEGs) when identifying potential pharmacological targets and improving precision therapeutics. The authors review existing literature, data, and computational methods dedicated to recognizing csEGs, and adapt these methods to predict csEGs in the context of kidney tissue. They experiment with four different prediction methods (XGEP, EPGAT, DeepHE and GrEGs) , using labels provided by four different identification approaches (CS0-CSX, OGEE, ADaM and FiPer). the paper highlights the importance of considering different validation techniques and the impact of data normalization and threshold settings on csEG identification. It also discusses the limitations of the methods, such as the lack of replicates and variations in read count distributions. Overall, the novelty of this paper lies in its comprehensive analysis and comparison of different methods for csEG identification in human kidney-specific cell lines, providing insights into the approaches used and the results obtained.
The pitfall of this manuscript is that it lacks replicates, which can lead to a large variance of the model. Moreover, different thresholds can be set to distinguish between essential genes (EGs) and non-essential genes (nEGs) from the final scores, resulting in different sets of EGs even when starting from the same initial dataset.
Comments:
1. It is interesting to see how the embedding from the GeneFormer Foundation model can benefit scEG prediction (https://doi.org/10.1038/s41586-023-06139-9)
2. The authors stated that their work does not involve benchmarking different models. However, I was wondering if they could potentially test a few more context-specific datasets beyond kidney tissues and then compare them to non-context-specific datasets. This would be particularly interesting since the authors have highlighted the significance of taking into account context-specific essential genes (csEGs).
Reviewer 2 Report
Comments and Suggestions for Authors
Giordano et al review the genetic concept of gene essentiality with particular focus on context-specific essential genes (csEGs). The authors provide a theoretical and experimental overview of the literature, data and computational methods dedicated to recognizing csEGs. Next, the authors applied four existing different prediction methods to kidney tissue.
I enjoyed reading the review sections (1-4). They were well written and informative.
The analysis section (5) can be improved. The authors claim to identify human kidney-specific core fitness genes (CFGs) by merging multiple methods. However, it is not clear whether the authors removed any common fitness genes. Without removing common fitness genes, the kidney CFGs are not truly specific. In addition, the authors should expand their analysis to a minimum of two tissues to focus on the context-specificity by contrasting two different tissues.
I commend the authors for making their code available.
Reviewer 3 Report
Comments and Suggestions for Authors
Giordano et al., touch on an interesting topic of essential genes which provide useful information that can be exploited in the development of personalized medicine aiming towards better health. The manuscript is comprehensive and well-written. However, looking at the overall structure and content, the manuscript seems more of a review than a research article. The manuscript can be accepted after these minor issues are resolved.
I suggest changing the article type to a review, which seems more suitable for this manuscript.
Figure 3: The legend position on the first plot is out of place compared to the other three. It’s better to have just one legend for all four plots. It can be placed either on top or below the x-axis.
Also, provide titles for the x- and y-axis in Figure 3.
Table A2: Metrics FPR and FNR are not used in the manuscript, so they should be removed.
Round 2
Reviewer 1 Report
Comments and Suggestions for Authors
No further comments
Reviewer 2 Report
Comments and Suggestions for Authors
The authors addressed my concerns.